# Comparison of Broth Microdilution, Disk Diffusion and Strip Test Methods for Cefiderocol Antimicrobial Susceptibility Testing on KPC-Producing *Klebsiella pneumoniae*

**DOI:** 10.3390/antibiotics12030614

**Published:** 2023-03-20

**Authors:** Federica Bovo, Tiziana Lazzarotto, Simone Ambretti, Paolo Gaibani

**Affiliations:** Microbiology Operative Unit, IRCCS Azienda Ospedaliero-Universitaria di Bologna, 40138 Bologna, Italy

**Keywords:** cefiderocol, antimicrobial susceptibility testing (AST), antimicrobial resistance (AMR), *Klebsiella pneumoniae* carbapenemase

## Abstract

The aim of this study was to compare the reference broth microdilution (BMD) method with the Disk Diffusion (DD) test and strip test against a collection of 75 well-characterized *Klebsiella pneumoniae* carbapenemase (KPC)-producing *Klebsiella pneumoniae* (KPC-Kp) clinical strains to assess cefiderocol (CFD) antimicrobial activity. Whole-genome sequencing was performed on KPC-Kp strains by Illumina iSeq100 platform. The Categorical Agreement (CA) between the BMD method and DD test was 92% (69/75) with a Major Error (ME) of 16.7% (6/36). Additionally, the CA between the BMD method and test strip was 90.7% (68/75) with a Very Major Error (VME) of 17.9% (7/39) and 82.7% (62/75) between the strip test and DD with a ME of 30.2%. KPC-Kp strains showing resistance to CFD were 27 out of 75 (36%) by three methods. Specifically, 51.9% (14/27) of KPC-Kp resistant to CFD harbored *bla*_KPC-3_, while 48.1% (13/27) harbored mutated *bla*_KPC-3_. Moreover, KPC-Kp strains carrying a mutated *bla*_KPC-3_ gene exhibited high MIC values (*p* value < 0.001) compared to wild-type *bla*_KPC-3_. In conclusion, the DD test resulted as a valid alternative to the BMD method to determine the in vitro susceptibility to CFD, while the strip test exhibited major limitations.

## 1. Introduction

Infections caused by carbapenem-resistant Enterobacteriales (CRE) organisms have become one of the greatest threats to public health over the past decade [1,2]. CRE-related infections can often lead to the development of bloodstream infections (BSIs), ventilator-associated pneumonia (VAP), intra-abdominal abscesses (IAAs) and urinary tract infections (UTIs) in hospitalized patients [1] and are often associated with high mortality rates in hospitals [1,2]. Resistance to β-lactams in CRE is due to different mechanisms, including β-lactamase production, porin loss and upregulation of efflux pumps [3]. In particular, among β-lactamase enzymes, Klebsiella pneumoniae carbapenemase (KPC) is the most common resistance mechanism in various countries [4]. KPC-producing bacteria are able to hydrolyze efficiently all β-lactams and often carry different other antimicrobial resistance determinants, thus resulting in a multidrug resistant (MDR) phenotype with limited available antimicrobial options [4]. To overcome these limited antimicrobial armamentaria for clinicians, new combinations of β-lactam/β-lactamase-inhibitors (βL-βLICs), such as ceftazidime-avibactam (CAZ-AVI), meropenem-vaborbactam (MER-VAB) and imipenem/cilastatin-relebactam (IMI-REL), have been developed since 2015 [5,6]. However, none of the new combinations of βL-βLICs are stable against metallo-β-lactamases [5]. At the same time, emerging resistance against these novel drug combinations has been recently reported [6,7].

In order to overcome these limitations, cefiderocol (CFD), formerly S-649266, an intravenously infused siderophoric cephalosporin [5,8], was approved by the US Food and Drug Administration (FDA) in 2019 for the treatment of UTIs and in 2020 for the treatment of bacterial hospital-acquired pneumonia (HAP) and VAP [2]. Subsequently, it was approved in Europe for the treatment of infections caused by gram-negative bacteria in adults with limited treatment options [5,9].

CFD exhibited structural stability against most β-lactamase resistance mechanisms of gram-negative bacteria [2,10], since it is able to enter the periplasmic space through the active iron transport system, as well as through passive diffusion through the porin channels of the outer membrane [5,8,9,10]. Once inside, it dissociates from iron, and the cephalosporin moiety binds primarily to penicillin-binding protein 3 (PBP3), inhibiting bacterial cell wall synthesis [2]. CFD revealed high in vitro bactericidal activity against *K. pneumoniae*, *Escherichia coli*, *Pseudomonas aeruginosa* and *Acinetobacter baumannii* [9] and showed intrinsic structural stability against most gram-negative bacteria producing β-lactamases, such as KPC [11], oxacillin carbapenemase (OXA), New Delhi metallo-β-lactamase (NDM) and Verona integron metallo-β-lactamase (VIM) [3,10]. However, in vitro sensitivity procedures used to evaluate this molecule against MDR microorganisms have wide limitations, due to its demanding preparation procedure, which includes iron-depleted cation-adjusted Mueller–Hinton broth (ID-CAMHB).

Hence, the aim of this work is to compare the broth microdilution reference method (BMD) currently used to assess CFD antimicrobial activity against KPC-producing Enterobacteriales and disk diffusion (DD) test and strip test sensitivity methods, describing the degree of concordance (Categorical Agreement) and errors (Very Major Error, Major Error and Minor Error) observed between the three different techniques analyzed.

## 2. Results

In this study, we selected 75 strains of KPC-producing K. pneumoniae (KPC-Kp) genotypically characterized by WGS.

The phenotypic profiles of KPC-Kp strains showed that 32 out of 75 (42.6%) KPC-Kp strains included in this study were resistant to CAZ-AVI and/or MER-VAB and/or IMI-REL. In particular, 10 out of 32 (31.3%) KPC-Kp strains were resistant only to CAZ-AVI, while five out of 32 (15.6%) were resistant only to MER-VAB, and one (3.1%) KPC-Kp strain was resistant exclusively to IMI-REL. Deeper analysis of KPC-Kp strains resistant to novel βL-βLICs showed that 21.9% (7/32) exhibited cross-resistance to CAZ-AVI, MER-VAB and IMI-REL; 12.5% (4/32) showed cross-resistance to CAZ-AVI and MER-VAB; 12.5% (4/32) showed cross-resistance to MER-VAB and IMI-REL; and 3.1% (1/32) showed cross-resistance to CAZ-AVI and IMI-REL.

Genotypic analysis revealed that 46.9% (15/32) of strains resistant to CAZ-AVI and/or MER-VAB and/or IMI-REL harbored *bla*_KPC-3_; 50% (16/32) harbored a mutated *bla*_KPC-3_ gene; and 3.1% (1/32) carried *bla*_KPC-2_. MLST analysis showed that the KPC-Kp strains included in this study that were resistant to CAZ-AVI and/or MER-VAB and/or IMI-REL belonged to different sequence types (ST) including ST101, ST258, ST307, ST512 and ST1519.

According to the EUCAST interpretative criteria, antimicrobial susceptibility testing (AST) performed with the BMD method on the 75 selected KPC-Kp strains included in this study exhibited MICs ranging between 0.25 and >16 mg/L. KPC-Kp strains CFD-susceptible (MIC ≤ 2 mg/L) were 48% (36/75), while 52% (39/75) of KPC-Kp strains were resistant to CFD (Figure 1). Applying the EUCAST interpretative criteria, the Categorical Agreement (CA) between the BMD method and DD test was 92% (69/75), and the Major Error (ME) was 16.7% (6/36). Very Major Error (VME) was not detected. Excluding the discordant strains within the Area of Technical Uncertainty (ATU), the degree of concordance become 98.6% (69/70), while the ME drops to 2.8% (1/36) (Figure 2A). The degree of agreement between the MIC values obtained from BMD method compared to strip test was 90.7% (68/75). VMEs were found in 17.9% (7/39), while Major Errors were not detected (Figure 2B). Finally, the degree of agreement between MIC observed from strip test and DD test was 82.7% (62/75), showing a ME of 30.2% (13/43). No VME was detected (Figure 2C).

KPC-Kp strains showing resistance to CFD by the BMD, DD and strip test methods were 27 out of 75 (36%) with a median MIC for CFD equal to 8 (IQR 32–4). Interestingly, genetic analysis showed that 14 out of 27 (51.9%) KPC-Kp strains resistant to CFD harbored a wild-type *bla*_KPC-3_ and exhibited a median MIC for CFD of 8 (IQR 4–8). Conversely, 13 out of 27 (48.1%) KPC-Kp resistant to CFD harbored a mutated *bla*_KPC-3_ gene and exhibited a median MIC for CFD equal to 32 (IQR 8–32).

MLST analysis performed on CFD-resistant KPC-Kp showed that ST512 (*n* = 19) was the most represented Sequence Type followed by ST307 (*n* = 5), ST1519 (*n* = 2) and ST101 (*n* = 1). Comparing MIC values between mutated and wild-type *bla*_KPC-3_ showed that KPC-Kp strains carrying mutation within Ω-loop exhibited high MIC values for CFD (*p* value < 0.001). It is noteworthy that 12 out of 27 KPC-Kp strains resistant to CFD harbored a mutated *bla*_KPC_ and showed cross-resistance to CAZ-AVI within a median MIC of 144 (IQR 112–256). Specifically, 58.3% (7/12) carried the *bla*_KPC-31_ variant. Furthermore, 15 out of 27 KPC-Kp strains displaying resistance to CFD were susceptible to CAZ-AVI within a median MIC of 4 (IQR 4–8). In particular, 14 out of 15 carried the wild-type *bla*_KPC-3_ gene, while one KPC-Kp strain carried the *bla*_KPC-85_ gene (Table 1). Interestingly, KPC-Kp harboring the mutated *bla*_KPC-3_ gene exhibited higher MIC values for CAZ-AVI than strains carrying the wild-type *bla*_KPC-3_ gene (Appendix A).

## 3. Discussion

Currently, CFD represents an effective antimicrobial agent for the treatment of infections caused by gram-negative bacteria in adults with limited treatment options available [5]. In particular, CFD is effective against gram-negative bacteria exhibiting resistance to carbapenem and could be considered for the treatment of difficult-to-treat (DTR) infections [12,13]. Although the BMD method is recommended by EUCAST as the reference method for the in vitro susceptibility assessment of CFD, other suitable and less laborious methods are available, such as the DD test and the strip test. Therefore, in this study we evaluated the susceptibility of CFD against 75 clinical KPC-Kp strains by comparing the accuracy of the DD test and MIC test strip to BMD method.

According to EUCAST criteria, the DD test compared to the reference method showed a degree of concordance of 92% and a ME of 16.7%. VME was not detected, thus suggesting that the DD method is more reliable in avoiding false negatives results. In addition, excluding values within the ATU range set by EUCAST (18–22 mm), the categorical agreement rises to 98.6%, and the ME drops to 2.8%, positioning the DD test as a notable alternative to the BMD method. However, particular attention must be placed on strains that fall inside the ATU range, which should be retested by the reference method, reported as resistant or ignored [10]. Moreover, potentially susceptible strains, showing a MIC of 2 mg/L but a DD test value between 18 and 22 mm, can be erroneously classified as resistant [10]. Considering the limited treatments options for patients affected by MDR pathogens or unable to tolerate aggressive drug regimens due to renal insufficiency, this could lead to negative implications for treatment of patients [10]. The comparison between the BMD and MIC test strip results showed a good degree of agreement. However, the MIC test strip incorrectly classified the BMD method-resistant isolates as CFD sensitive (i.e., VME). In particular, following the EUCAST interpretative criteria, the CA was 90.7%, and the VME 17.9%. Previous studies showed that KPC-Kp strains harboring mutations within the Ω-loop of the KPC-3 enzyme can exhibit increased MIC values for CFD and resistance to CAZ-AVI [11,14,15]. Indeed, mutations in the *bla*_KPC-3_ gene are likely responsible for extending the resistance spectrum by increasing CFD MIC values. In agreement with previous studies, our results revealed that strains harboring mutated *bla*_KPC-3_ showed higher MICs for CFD and acquisition of resistance to CAZ-AVI, compared to wild type *bla*_KPC-3_. In this context, further analyses are needed in order to better understand the emergence of resistance mechanisms, which can lead to the development of cross-resistance to the latest generation of antibiotics in KPC-Kp strains.

In conclusion, our results demonstrated that the DD test is a valid alternative for determining in vitro susceptibility to CFD, since the BMD method has proved to be challenging for routine laboratory analysis, especially for those laboratories managing large numbers of patients. The ATU values can be used as an indication of uncertainty in the test procedure and therefore as an index of difficulty in interpreting results. In these cases, it may be useful to subsequently confirm uncertain MIC values with the reference method, in order to have reliable results.

## 4. Materials and Methods

### 4.1. Phenotypic Analysis

*K. pneumoniae* strains included in the study were isolated from patients admitted at IRCCS Policlinico di Sant’Orsola, a 1.420-bed university hospital with an average of 72,000 admissions per year. Bacterial strains were identified by MALDI-TOF MS assay (Bruker Daltonics, Bremen, Germany). AST was performed using the MicroScan Walkaway system. MICs for novel βL-βLICs were confirmed with MIC test strips (Liofilchem, Roseto degli Abruzzi, Italy). Carbapenemase production was detected by NG-Test CARBA 5 (NG Biotech, France) and confirmed with molecular assay Xpert Carba-R (Cepheid, Sunnyvale, CA, USA). AST for cefiderocol was performed using broth microdilution with ID-CAMHB. DD and MIC test strips (Liofilchem, Italy) were used on regular non-supplemented Mueller–Hinton agar (Liofilchem, Italy). MIC results were interpreted following the European Committee on Antimicrobial Susceptibility Testing (EUCAST) clinical breakpoints v12.0 (accessed on 10 January 2023 at: https://www.eucast.org/clinical_breakpoints/) [16].

### 4.2. Broth Microdilution (BMD)

AST for CFD was performed removing iron from Muller–Hinton broth using Chelex^®^100 binding resin (Bio-Rad Laboratories, CA, USA). The broth was then filtered through a 0.22 μm pore size filter. CaCl_2_ (22.5 mg/L), MgCl_2_ (11.2 mg/L) and ZnSO_4_ (5.6 mg/L) were supplemented to the depleted broth. Cefiderocol powder (Shionogi & Co., Ltd., Osaka, Japan) was dissolved in sterile saline solution (320 mg/L). Strains were inoculated at a concentration of 1.5 × 10^8^ CFU/mL and incubated for 20 h at 35 °C [17,18]. MIC was determined as the first well in which the reduction of growth corresponded to a button of <1 mm, and results were interpreted using EUCAST clinical breakpoints [19].

### 4.3. Genomic Analysis

Genomic DNA was extracted from purified cultures of *K. pneumoniae* using DNeasy Blood&Tissue Kit (Qiagen, Basel, Switzerland) by following the manufacturer’s instructions and further cleaned up with AMPure XP magnetic beads (Beckman Coulter). Whole genome analysis was performed as previously described [20]. Briefly, bacterial genomes libraries were prepared using Illumina DNA Prep paired-end kit and sequenced by the Illumina iSeq 100 platform (iSeq Reagent Kit v2, Illumina, San Diego, CA, USA) using iSeq Reagent kit v2 with 2 × 150 paired-end reads. Read sets were evaluated for sequence quality and read-pair length using FastQC software (available at: https://www.bioinformatics.babraham.ac.uk/projects/fastqc/, accessed on 10 January 2023). Genome assemblies were performed using with SPAdes v.3.10 with careful settings and were polished with Pilon v.1.23. Annotation was automatically carried out using the RAST server (available at: https://rast.nmpdr.org, accessed on 10 January 2023) and manually curated using Artemis v.17.0.1. Antimicrobial resistance genes and MLST analysis were assessed using the online platform (available at: https://cge.cbs.dtu.dk/services/, accessed on 10 January 2023). β-lactamase content was confirmed by using BLAST analysis against CARDB and Beta-Lactamase-Database (http://bldb.eu, accessed on 10 January 2023).

## Figures and Tables

**Figure 1 antibiotics-12-00614-f001:**
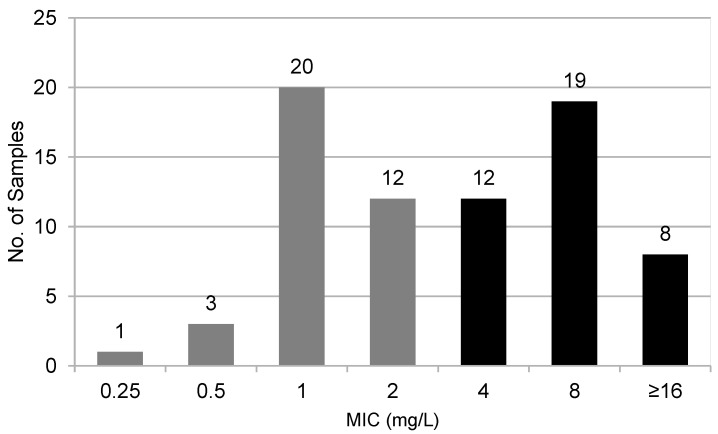
Distribution of the 75 KPC-Kp strains MIC values using the BMD method according to EUCAST clinical breakpoints: gray columns indicate strains showing sensitive MIC values (≤2 mg/L); black columns indicate strains showing resistant MIC values (>2 mg/L).

**Figure 2 antibiotics-12-00614-f002:**
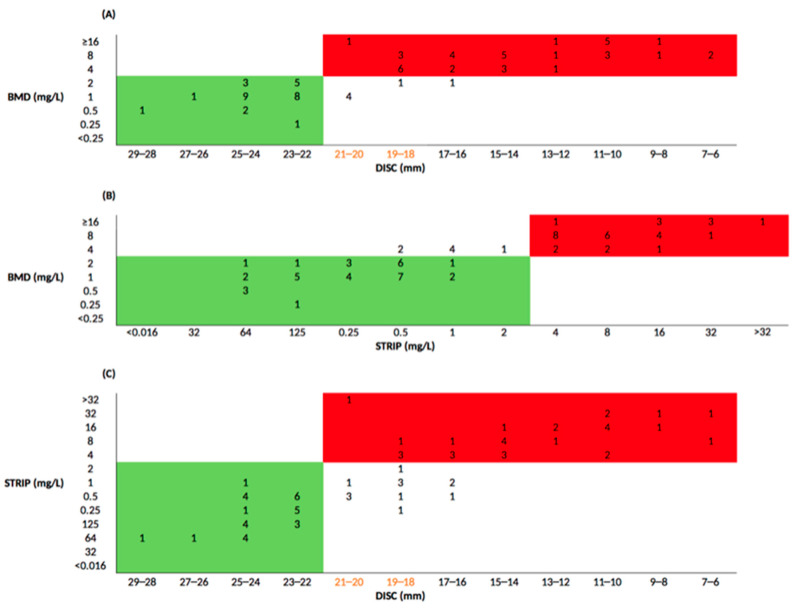
Comparison between AST methods applying EUCAST clinical breakpoints. Green and red boxes highlight the number of KPC-Kp strains sensitive and resistant to both methods, respectively; yellownumbers indicate strains within the ATU zone. (**A**) Comparison between BMD and DD MIC values; (**B**) Comparison between BMD and strip test MIC values; (**C**) Comparison between strip test and DD MIC values.

**Table 1 antibiotics-12-00614-t001:** Phenotypic and genotypic characteristics of KPC-Kp strains resistant to CFD included in this study. Reduced susceptibility to antimicrobial molecules are indicated in bold.

SAMPLES	KPC	ST ^b^	MIC (mg/L) ^a^					
			CFD ^c^			CAZ-AVI ^f^	MER-VAB ^g^	IMI-REL ^h^
			BMD ^d^	DD ^e^	STRIP			
BOT-CAR	KPC-31	1519	**8**	**15**	**4**	**>256**	2	0.25
KpBO11	KPC-3	512	**4**	**13**	**8**	8	**256**	1
KpBO12	KPC-3	512	**4**	**15**	**4**	8	**256**	0.5
KpBO30	KPC-31	307	**8**	**16**	**4**	**256**	0.125	0.125
KpBO32	KPC-31	307	**4**	**14**	**8**	**256**	1	0.5
CNTD42	KPC-3	512	**8**	**15**	**4**	8	0.5	0.5
BAT15	KPC-3	512	**8**	**11**	**4**	4	4	1
BAT13	KPC-3	512	**8**	**17**	**8**	4	0.25	0.25
CAZ03	KPC-31	1519	**≥16**	**10**	**32**	**>256**	8	**4**
CAZ30	KPC-31	307	**≥16**	**13**	**32**	**>256**	0.047	0.19
CAZ42	KPC-3	512	**8**	**17**	**4**	8	8	1
CAZ51	KPC-3	512	**8**	**15**	**8**	8	8	1
BAT32	KPC-3	307	**4**	**14**	**16**	4	0.25	0.5
BAT33	KPC-3	307	**8**	**10**	**16**	≤2	0.064	0.25
CAZ147	KPC-31	101	**≥16**	**9**	**32**	**256**	2	1
BAT142	KPC-3	512	**≥16**	**11**	**4**	4	1.5	1.5
BO784	KPC-66	512	**16**	**10**	**32**	**32**	**16**	**4**
BO830	KPC-68	512	**16**	**10**	**32**	**64**	**48**	**4**
BO739	KPC-3	512	**8**	**12**	**16**	8	**32**	**4**
BO714	KPC-125	512	**8**	**11**	**32**	**>256**	**16**	**4**
BO743	KPC-121	512	**8**	**9**	**16**	**>256**	**24**	**8**
BO837	KPC-3	512	**8**	**7**	**8**	4	**32**	**4**
BO999	KPC-31	512	**8**	**7**	**32**	**>256**	**32**	**4**
BAT146	KPC-3	512	**8**	**15**	**8**	4	**32**	**4**
BAT147	KPC-3	512	**8**	**17**	**4**	4	0.5	0.75
TO6	KPC-49	512	**≥16**	**10**	**16**	**32**	1.5	0.5
BAT154	KPC-85	512	**8**	**14**	**8**	8	2	0.75

^a^ Applying EUCAST breakpoint; ^b^ ST, Sequence Type; ^c^ CFD, cefiderocol; ^d^ BMD, broth microdilution; ^e^ DD, disk diffusion (inhibition zone diameter in mm); ^f^ CAZ-AVI, ceftazidime-avibactam; ^g^ MER-VAB, meropenem-vaborbactam; ^h^ IMI-REL, imipenem-relebactam.

## Data Availability

Not applicable.

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
