# Peer review of "Comparison of Broth Microdilution, Disk Diffusion and Strip Test Methods for Cefiderocol Antimicrobial Susceptibility Testing on KPC-Producing Klebsiella pneumoniae"

_antibiotics, 2023, doi:10.3390/antibiotics12030614_

Round 1

Reviewer 1 Report

Carbapenem-resistant Enterobacteriales (CRE) organisms are one of the greatest threats to public health of humans. Broth microdilution (BMD), Disk Diffusion (DD) test, and Strip test were the methods used for antimicrobial susceptibility of bacteria. However, the consistence is yet to be compared. In this study, the authors compared broth microdilution, disk diffusion and strip test methods for Cefiderocol antimicrobial susceptibility testing on KPC-producing Klebsiella pneumoniae. This work is significant for antibiotics implications. Some revisions are needed as followed.

1.      K. pneumoniae is italic, please check all.

2.      After first time, the other Klebsiella pneumoniae should be K. pneumoniae.

3.      ST101, ST258, ST307, ST512, ST1519, and is added before the last one.

4.      In Figure 1 all the letters are capital, please change it.

5.      In Figure 2, please improve the resolution of this picture.

6.      in vitro is italic.

Author Response

Reviewer 1

Carbapenem-resistant Enterobacteriales (CRE) organisms are one of the greatest threats to public health of humans. Broth microdilution (BMD), Disk Diffusion (DD) test, and Strip test were the methods used for antimicrobial susceptibility of bacteria. However, the consistence is yet to be compared. In this study, the authors compared broth microdilution, disk diffusion and strip test methods for Cefiderocol antimicrobial susceptibility testing on KPC-producing Klebsiella pneumoniae. This work is significant for antibiotics implications. Some revisions are needed as followed. 

Authors’ reply and amendments: We thanks reviewer for his comment. We revised the manuscript accordingly with the reviewer suggestions

1.      K. pneumoniae is italic, please check all. 

Authors’ reply and amendments:   the term was modified as requested

2.      After first time, the other Klebsiella pneumoniae should be K. pneumoniae.

Authors’ reply and amendments:   the term was modified as requested

  1. ST101, ST258, ST307, ST512, ST1519, and is added before the last one. 

Authors’ reply and amendments:   the STs were modified as requested    

  1. In Figure 1 all the letters are capital, please change it.

Authors’ reply and amendments:   the term was modified as requested

  1. In Figure 2, please improve the resolution of this picture.

Authors’ reply and amendments:   the quality of the figure was improved

 as requested

  1. in vitro is italic.

Authors’ reply and amendments:   the term was modified as requested        

Reviewer 2 Report

Congrats for this study , very intersting the rates of resistant to Cefedericol.

I suggest mentioning and maybe compare with a table  the results using the CLSI breakpoints , how much chnage the results.

Reviewer 3 Report

Summary:  In the manuscript by Bovo et al., the authors compare broth microdilution, disk diffusion, and MIC test strip methods to assess susceptibility testing of carbapenem-resistant K. pneumoniae strains to a relatively new antibiotic, Cefiderocol (CFD).  Currently, broth microdilution is used to assess susceptibility to CFD.  In this study, the authors compare results of broth microdilution to disk diffusion and MIC strip tests.  Using EUCAST guidelines for Categorical Agreement, the authors report that disk diffusion is a valid alternative to the broth dilution.  One of the strengths of this study is the MLST of the K. pneumoniae strains, allowing readers to assess susceptibility based on sequence type.  This information will be of interest to those interested in rapid methods to assess gram-negative susceptibility to CFD.

Review:  The information is relevant to those working with carbapenem-resistant gram-negative bacteria and treatment options in patients.  It provides information about valid alternative sensitivity testing for a relatively new antibiotic.  Because the antibiotic is relatively new, the authors provide only twenty references; however, references could be added that show how others have used similar comparisons with other antibiotics. 

Major Comments:

English language and grammar editing is required.

The use of "disk" for disk diffusion varies from "disk" to "disc" throughout the manuscript.

KPC should be defined in the abstract.

Food and Drug Administration should be US Food and Drug Administration.

The first sentence of paragraph 3 in the Introduction is too similar in wording to a sentence in reference 9 with the substitutions of "overtake" for "overcome."  However, "overtake" is not appropriate in reference to resistance mechanisms.  Suggest re-wording the sentence appropriately.

Results:  

The sentence beginning at the end of Line 4 with "In particular, 10 out of..." is unclear.  There are extra/missing words.

Paragraph 4:  This is the first use the abbreviation AST in the manuscript.  AST was defined in the M&M, but that is at the end of manuscript.  Define here.

Also in paragraph 4, authors switch between present and past tense.  Should all be past tense (example, "was" not "is"). 

Figure 1:   Remove "Broth Microdilution" title from the table.  

X axis:  put units next to MIC (mg/L)

Y axis:  a percentage would provide a better depiction.  If staying with number of samples, please spell out instead of using "N. Samples" and provide the total number of samples in the figure legend.

Figure 2:  move the axis titles to the centers of the figures and provide the units for the X and Y axis values - example BMD (mg/L).  

Also suggest changing (a), (b), (c) to A, B, C and moving numbers to top edge of each figure (instead of having axis title at top) so that it is more visible to the reader.

Figure 2 legend:  Is this really a "proportion"?  Should it not just read "In green and red, the numbers of isolates..."  

The figure legend is also misleading with the "yellow-ish" numbers?.  Why not just highlight the yellow zone on the figure?  Otherwise, make the Figure legend clearer:  example - "Green and red boxes highlight the numbers of isolates...to both methods.  Yellow boxes/Yellow numbers indicate isolates within the ATU zone."

Discussion - Line 1 - suggest "approved" instead of "valid"

Line 2 - suggest "...limited treatment options available"

Paragraph 2, Line 5 - suggest "positioning" instead of "posing"

Author Response

Comments and Suggestions for Authors

Summary:  In the manuscript by Bovo et al., the authors compare broth microdilution, disk diffusion, and MIC test strip methods to assess susceptibility testing of carbapenem-resistant K. pneumoniae strains to a relatively new antibiotic, Cefiderocol (CFD).  Currently, broth microdilution is used to assess susceptibility to CFD.  In this study, the authors compare results of broth microdilution to disk diffusion and MIC strip tests.  Using EUCAST guidelines for Categorical Agreement, the authors report that disk diffusion is a valid alternative to the broth dilution.  One of the strengths of this study is the MLST of the K. pneumoniae strains, allowing readers to assess susceptibility based on sequence type.  This information will be of interest to those interested in rapid methods to assess gram-negative susceptibility to CFD.

Review:  The information is relevant to those working with carbapenem-resistant gram-negative bacteria and treatment options in patients.  It provides information about valid alternative sensitivity testing for a relatively new antibiotic.  Because the antibiotic is relatively new, the authors provide only twenty references; however, references could be added that show how others have used similar comparisons with other antibiotics.

Authors’ reply and amendments: We thanks reviewer for his comment. Based on the new data reported the literature is limited about new antibiotics in relation to the diagnostic tests. However, we added the most important references in this field.

Major Comments:

English language and grammar editing is required.

Authors’ reply and amendments: the manuscript was checked for English as requested

The use of "disk" for disk diffusion varies from "disk" to "disc" throughout the manuscript.

Authors’ reply and amendments: the sentence was modified as requested

KPC should be defined in the abstract.

Authors’ reply and amendments: the term was modified as requested

Food and Drug Administration should be US Food and Drug Administration.

Authors’ reply and amendments: the term was modified as requested

The first sentence of paragraph 3 in the Introduction is too similar in wording to a sentence in reference 9 with the substitutions of "overtake" for "overcome."  However, "overtake" is not appropriate in reference to resistance mechanisms.  Suggest re-wording the sentence appropriately. Authors’ reply and amendments: the sentence wasmodified as requested

Results:  

The sentence beginning at the end of Line 4 with "In particular, 10 out of..." is unclear.  There are extra/missing words.

Authors’ reply and amendments: the sentence was modified as requested

Paragraph 4:  This is the first use the abbreviation AST in the manuscript.  AST was defined in the M&M, but that is at the end of manuscript.  Define here.

Authors’ reply and amendments: the term was explained as requested

Also in paragraph 4, authors switch between present and past tense.  Should all be past tense (example, "was" not "is").

Authors’ reply and amendments: the sentence was modified as requested

Figure 1:   Remove "Broth Microdilution" title from the table.  

Authors’ reply and amendments: the sentence was modified as requested

X axis:  put units next to MIC (mg/L)

Authors’ reply and amendments: the units for X axis was added as requested

Y axis:  a percentage would provide a better depiction.  If staying with number of samples, please spell out instead of using "N. Samples" and provide the total number of samples in the figure legend.

Authors’ reply and amendments: the percentage was added as requested

Figure 2:  move the axis titles to the centers of the figures and provide the units for the X and Y axis values - example BMD (mg/L).  

Authors’ reply and amendments: the figure 2 was modified as requested

Also suggest changing (a), (b), (c) to A, B, C and moving numbers to top edge of each figure (instead of having axis title at top) so that it is more visible to the reader.

Authors’ reply and amendments: the figure was modified as requested

Figure 2 legend:  Is this really a "proportion"?  Should it not just read "In green and red, the numbers of isolates..."  

Authors’ reply and amendments: the sentence was modified as requested

The figure legend is also misleading with the "yellow-ish" numbers?.  Why not just highlight the yellow zone on the figure?  Otherwise, make the Figure legend clearer:  example - "Green and red boxes highlight the numbers of isolates...to both methods.  Yellow boxes/Yellow numbers indicate isolates within the ATU zone."

Authors’ reply and amendments: the figure was modified as requested

Discussion - Line 1 - suggest "approved" instead of "valid"

Authors’ reply and amendments: the sentence was modified as requested

Line 2 - suggest "...limited treatment options available"

Authors’ reply and amendments: the sentence was modified as requested

Paragraph 2, Line 5 - suggest "positioning" instead of "posing"

Authors’ reply and amendments: the term was modified as requested

Reviewer 4 Report

The study presents "Comparison of broth microdilution, disk diffusion and strip test methods for Cefiderocol antimicrobial susceptibility testing on KPC-producing Klebsiella pneumoniae". The study is interesting, however, there are few minor and major issues and suggestions to be corrected toward the final version of your review.

General: Submit the revised version with continuous line numbers which will be easy for reviewers. Please download the highlighted version of the paper. I have highlighted and commented in yellow. All the bacterial and genes names should be in italic. 

Introduction:

Second line of the introduction "The word "led" should be replaced with "lead" to make it grammatically correct."

3rd line "This sentence need to be rephrase. Delete the repetition. Better to write ""Resistance to β-lactams in CRE is due to different mechanisms, including β-lactamase production, porin loss, and upregulation of efflux pump"

4th line "Please correct the sentence. "K. pneumoniae carbapenemasi (KPC) is the most common resistance mechanism in different countries [4]." - it would be better to say "K. pneumoniae carbapenemase (KPC) is the most common resistance mechanism in various countries [4]."

5th line "It should be "KPC-producing bacteria"  "carried" Write the correct verb. I think it should be "carry"

Last sentence of first para "This sentence has a minor error in subject-verb agreement. Please correct "

Results:

In the first sentence, "a collection of 75 strains" should be "75 strains" (the "a collection of" is unnecessary).

Second sentence ""KPC-Kp" should be "KPC-producing Klebsiella pneumoniae" for clarity. "phenotypic profile of KPC-Kp isolates showed" should be "the phenotypic profiles of KPC-Kp isolates showed" to match the plural subject."

In the sixth sentence of result  section, "KPC-Kp resistant to novel βL-βLICs" should be "KPC-Kp strains resistant to novel βL-βLICs...........

In the eighth sentence, "KPC-Kp strains resistant to CAZ-AVI and/or MER-VAB and/or IMI-REL included in this study" should be "the KPC-Kp strains included in this study that were resistant to CAZ-AVI and/or MER-VAB and/or IMI-REL" for clarity.

In the tenth sentence, "susceptible to CFD" should be "CFD-susceptible" for clarity.

Page 3 first sentence "Instead of "by the three methods," consider specifying what those methods were. For example, "by disk diffusion, Etest, and broth microdilution."

Page 3 3rd line "Instead of "On opposite," use "On the other hand" or "In contrast."

Page 3 2nd para second line "Instead of "Comparison of MIC values," use "Comparing MIC values."

Page 3 2nd para 3rd line "Instead of "Of note," consider rephrasing to "It is noteworthy that."

Discussion: 

Please check the manuscript with highlighted comments. As there is no line numbers in the manuscript.

First sentence "Instead of "valid antimicrobial choice," it might be clearer to say "effective antimicrobial agent."

"limited treatment available options" should be "limited treatment options available."

Methods

Also check the manuscript with comments

Author Response

Authors’ reply and amendments: We thanks reviewer for his comment. We revised the manuscript accordingly with the reviewer suggestions.

Introduction:

Second line of the introduction "The word "led" should be replaced with "lead" to make it grammatically correct."

Authors’ reply and amendments: the term was modified as requested

3rd line "This sentence need to be rephrase. Delete the repetition. Better to write ""Resistance to β-lactams in CRE is due to different mechanisms, including β-lactamase production, porin loss, and upregulation of efflux pump"

Authors’ reply and amendments: the sentence was modified as requested

4th line "Please correct the sentence. "K. pneumoniae carbapenemasi (KPC) is the most common resistance mechanism in different countries [4]." - it would be better to say "K. pneumoniae carbapenemase (KPC) is the most common resistance mechanism in various countries [4]."

5th line "It should be "KPC-producing bacteria"  "carried" Write the correct verb. I think it should be "carry"

Authors’ reply and amendments: the sentence was modified as requested

Last sentence of first para "This sentence has a minor error in subject-verb agreement. Please correct "

Authors’ reply and amendments: the sentence was modified as requested

Results:

In the first sentence, "a collection of 75 strains" should be "75 strains" (the "a collection of" is unnecessary).

Authors’ reply and amendments: the sentence was modified as requested

Second sentence ""KPC-Kp" should be "KPC-producing Klebsiella pneumoniae" for clarity.

Authors’ reply and amendments: the term was cited in full the first time and then abbreviated in the manuscript

"phenotypic profile of KPC-Kp isolates showed" should be "the phenotypic profiles of KPC-Kp isolates showed" to match the plural subject."

Authors’ reply and amendments: the sentence was modified as requested

In the sixth sentence of result  section, "KPC-Kp resistant to novel βL-βLICs" should be "KPC-Kp strains resistant to novel βL-βLICs........…

Authors’ reply and amendments: the sentence was modified as requested

In the eighth sentence, "KPC-Kp strains resistant to CAZ-AVI and/or MER-VAB and/or IMI-REL included in this study" should be "the KPC-Kp strains included in this study that were resistant to CAZ-AVI and/or MER-VAB and/or IMI-REL" for clarity.

Authors’ reply and amendments: the sentence was modified as requested

In the tenth sentence, "susceptible to CFD" should be "CFD-susceptible" for clarity.

Authors’ reply and amendments: the sentence was modified as requested

Page 3 first sentence "Instead of "by the three methods," consider specifying what those methods were. For example, "by disk diffusion, Etest, and broth microdilution."

Authors’ reply and amendments: the sentence was modified as requested

Page 3 3rd line "Instead of "On opposite," use "On the other hand" or "In contrast."

Authors’ reply and amendments: the terms were modified as requested

Page 3 2nd para second line "Instead of "Comparison of MIC values," use "Comparing MIC values."

Authors’ reply and amendments: the terms were modified as requested

Page 3 2nd para 3rd line "Instead of "Of note," consider rephrasing to "It is noteworthy that."

Authors’ reply and amendments: the sentence was modified as requested

Discussion: 

Please check the manuscript with highlighted comments. As there is no line numbers in the manuscript.

Authors’ reply and amendments: the manuscript was modified as requested

First sentence "Instead of "valid antimicrobial choice," it might be clearer to say "effective antimicrobial agent."

Authors’ reply and amendments: the sentence was modified as requested

"limited treatment available options" should be "limited treatment options available."

Authors’ reply and amendments: the sentence was modified as requested

Methods

Also check the manuscript with comments

Authors’ reply and amendments: the manuscript was fully checked as requested

Reviewer 5 Report

1-      Introduction, this sentence seems lack word’s’ ‘..Resistance to β-lactams in CRE are due to different including, including..’ Different what?.

2-      Please verify in all the manuscript, the name of bacteria should be written in italic.

3-      Introduction, correct ‘…K. pneumoniae carbapenemase (KPC) is the most common..’

4-      Sometimes you wrote “strains’ and sometimes ‘isolates; since the 75 K. pneumoniae were already characterized by WGS, you are allowed to use the term ‘strain’ better than the term ‘isolate’.

5-      Add the “Acknowledgments:’, you acknowledge who ?

6-      List of references, delete the number ‘1’ in the first reference.

7-      At the end of the article (after the list of references), please delete the word ‘Disclaimer/’

Author Response

Authors’ reply and amendments: We thanks reviewer for his comment. We revised the manuscript accordingly with the reviewer suggestions

1-   Introduction, this sentence seems lack word’s’ ‘..Resistance to β-lactams in CRE are due to different including, including..’ Different what?.

Authors’ reply and amendments: the sentence was modified as requested

2-   Please verify in all the manuscript, the name of bacteria should be written in italic.

Authors’ reply and amendments: the terms were checked as requested

3-   Introduction, correct ‘…K. pneumoniae carbapenemase (KPC) is the most common..’

Authors’ reply and amendments: the sentence was modified as requested

4-      Sometimes you wrote “strains’ and sometimes ‘isolates; since the 75 K. pneumoniae were already characterized by WGS, you are allowed to use the term ‘strain’ better than the term ‘isolate’.

Authors’ reply and amendments: the terms were checked as requested

5- Add the “Acknowledgments:’, you acknowledge who?

Authors’ reply and amendments: the sentence was modified as requested

6-   List of references, delete the number ‘1’ in the first reference.

Authors’ reply and amendments: the reference’s list was modified as requested

7-   At the end of the article (after the list of references), please delete the word ‘Disclaimer/’

Authors’ reply and amendments: the term was removed as requested

Reviewer 6 Report

Comments to the Author

This manuscript compares the reference broth microdilution method with the Disk Diffusion test and Strip test against Klebsiella pneumoniae carbapenemase (KPC)-producing clinical strains to assess cefiderocol antimicrobial activity. The authors’ findings are interesting and worthy of publication after consideration of the following:

1-Line numbering is essential

2-In Abstract  

“Aim of this study was to compare the reference broth microdilution (BMD) method with Disk Diffusion (DD) test and Strip test against a collection of 75 well-characterized KPC-producing Klebsiella pneumoniae (KPC-Kp) clinical strains.”:  should be “Aim of this study was to compare the reference broth microdilution (BMD) method with Disk Diffusion (DD) test and Strip test against a collection of 75 well-characterized Klebsiella pneumoniae carbapenemase (KPC)-producing Klebsiella pneumoniae (KPC-Kp) clinical strains to assess cefiderocol (CFD) antimicrobial activity.

bla KPC-3: “KPC-3”should be a lower case

3-Keywords: PLEASE ADD “Klebsiella pneumoniae carbapenemase”

4-In introduction “to different including, including”: delete “including”

K. pneumoniae carbapenemasi (KPC) : italicize “K. pneumoniae” and correct “carbapenemase”

5-Figure 1: please write on Y axis “No. of samples” instead of “N. SAMPLES”

6-Figure 2: Please write what Y and X axis indicates (e.g mg/L or mm)

7-Table 1: DD (inhibition zone diameter in mm) should be added to footnote

8-Discussion “the in vitro susceptibility” : italicize “the in vitro”

“inside the ATU range,” : [The Area of Technical Uncertainty (ATU)] write in full at the first mention

9-Materials and Methods

“Klebsiella pneumoniae isolates included in the study”: please write the total number

“was dissolved in sterile saline water (320 mg/L).”: sterile saline or water?

“Genomic DNA was extracted from purified cultures of Klebsiella pneumoniae”: K. pneumoniae (at first it was written in full then writes the abbreviation through the manuscript)

It is mandatory to write the bioproject number of the WGS data 

Author Response

Authors’ reply and amendments: We thanks reviewer for his comment. We revised the manuscript accordingly with the reviewer suggestions

The authors’ findings are interesting and worthy of publication after consideration of the following:

1-Line numbering is essential

Authors’ reply and amendments: the manuscript was modified as requested

2-In Abstract  

Aim of this study was to compare the reference broth microdilution (BMD) method with Disk Diffusion (DD) test and Strip test against a collection of 75 well-characterized KPC-producing Klebsiella pneumoniae (KPC-Kp) clinical strains.”:  should be “Aim of this study was to compare the reference broth microdilution (BMD) method with Disk Diffusion (DD) test and Strip test against a collection of 75 well-characterized Klebsiella pneumoniae carbapenemase (KPC)-producing Klebsiella pneumoniae (KPC-Kp) clinical strains to assess cefiderocol (CFD) antimicrobial activity.

Authors’ reply and amendments: the sentence was modified as requested

bla KPC-3: “KPC-3”should be a lower case.

Authors’ reply and amendments: the term was modified as requested

3-Keywords: PLEASE ADD “Klebsiella pneumoniae carbapenemase”

Authors’ reply and amendments: the term was added as requested

4-In introduction “to different including, including”: delete “including”

Authors’ reply and amendments: the sentence was modified as requested

K. pneumoniae carbapenemasi (KPC) : italicize “K. pneumoniae” and correct “carbapenemase”

Authors’ reply and amendments: the terms were modified as requested

5-Figure 1: please write on Y axis “No. of samples” instead of “N. SAMPLES”

Authors’ reply and amendments: the term was modified as requested

6-Figure 2: Please write what Y and X axis indicates (e.g. mg/L or mm)

Authors’ reply and amendments: the terms were added as requested

7-Table 1: DD (inhibition zone diameter in mm) should be added to footnote

Authors’ reply and amendments: the term was added as requested

8-Discussion “the in vitro susceptibility” : italicize “the in vitro”

inside the ATU range,” : [The Area of Technical Uncertainty (ATU)] write in full at the first mention

Authors’ reply and amendments: The first mention is written previous in full in the manuscript and then abbreviated, as requested

9-Materials and Methods

Klebsiella pneumoniae isolates included in the study”: please write the total number

was dissolved in sterile saline water (320 mg/L).”: sterile saline or water?

Authors’ reply and amendments: the sentence was modified as requested

Genomic DNA was extracted from purified cultures of Klebsiella pneumoniae”: K. pneumoniae (at first it was written in full then writes the abbreviation through the manuscript)

Authors’ reply and amendments: the sentence was modified as requested

 It is mandatory to write the bioproject number of the WGS data 

Authors’ reply and amendments: Most of the isolates are part of a bioproject under sequencing. Therefore, we cannot add the bioproject number for most of the isolates and considering the high number of strains included in this study we opted to not add the single accession number

Round 2

Reviewer 3 Report

There are still English grammar and punctuation corrections that need to be addressed, specifically the lack of appropriate articles (a, an, the) and comma usage.  

Examples:  L44 - there should not be a comma after "UTI"; L54 - there should be no comma after [9]; L56 - there should be a comma after (NDM), etc.

Suggested edits:

L10 - begin with "The aim..."

L31 - spell out Klebsiella the first time it is used in the main body (not just the abstract), then it can be abbreviated as K.

L36 - because "these" is used, armamentarium should be pluralized to armamentaria

L37 - should be an "and" after (MER-VAB),

L44 - should be UTIs as in L28

L48 - suggest "exhibited" instead of "revealed"

L81 - "type" should be "types"

LL90-94 - because the figures were changed to 2A, 2B, 2C, the text should be changed as well

L98 - suggest "Conversely," instead of "On the other hand,"

L110 - "harboring the..."

L154 - "the BMD...:

The references also need a consistent formatting.

Author Response

Comments and Suggestions for Authors

There are still English grammar and punctuation corrections that need to be addressed, specifically the lack of appropriate articles (a, an, the) and comma usage.

Authors’ reply and amendments: We thanks reviewer for his comment. We modified the manuscript was modified accordingly to the referee suggestions

Examples: L44 - there should not be a comma after "UTI"; L54 - there should be no comma after [9]; L56 - there should be a comma after (NDM), etc.

Authors’ reply and amendments: the sentence was modified as requested

Suggested edits:

L10 - begin with "The aim…"

Authors’ reply and amendments: the sentence was modified as requested

L31 - spell out Klebsiella the first time it is used in the main body (not just the abstract), then it can be abbreviated as K.

Authors’ reply and amendments: the term was modified as requested

L36 - because "these" is used, armamentarium should be pluralized to armamentaria

Authors’ reply and amendments: the term was modified as requested

L37 - should be an "and" after (MER-VAB),

Authors’ reply and amendments: the sentence was modified as requested

L44 - should be UTIs as in L28

Authors’ reply and amendments: the sentence was modified as requested

L48 - suggest "exhibited" instead of "revealed"

Authors’ reply and amendments: the term was modified as requested

L81 - "type" should be "types"

Authors’ reply and amendments: the term was modified as requested

LL90-94 - because the figures were changed to 2A, 2B, 2C, the text should be changed as well

Authors’ reply and amendments: the figures were modified as requested

L98 - suggest "Conversely," instead of "On the other hand,"

Authors’ reply and amendments: the sentence was modified as requested

L110 - "harboring the…"

Authors’ reply and amendments: the term was modified as requested

L154 - "the BMD...:

Authors’ reply and amendments: the term was modified as requested

The references also need a consistent formatting.

Authors’ reply and amendments: the references were modified as requested

Reviewer 4 Report

After carefully reviewing the revised manuscript, I am pleased to inform you that the authors have satisfactorily addressed all the concerns raised during the initial review process.

Author Response

After carefully reviewing the revised manuscript, I am pleased to inform you that the authors have satisfactorily addressed all the concerns raised during the initial review process.

Authors’ reply and amendments: We thanks reviewer for his comment. We revised the manuscript accordingly with the previous reviewer suggestions to improve the quality of the study.